# Problematic Internet Use among Polish Students: Prevalence, Relationship to Sociodemographic Data and Internet Usage Patterns

**DOI:** 10.3390/ijerph20032434

**Published:** 2023-01-30

**Authors:** Marta Kożybka, Iwona Radlińska, Marcin Kolwitz, Beata Karakiewicz

**Affiliations:** 1Subdepartment of Medical Law, Department of Social Medicine, Faculty of Health Sciences, Pomeranian Medical University in Szczecin, ul. Żołnierska 48, 71-210 Szczecin, Poland; 2Subdepartment of Social Medicine and Public Health, Department of Social Medicine, Pomeranian Medical University in Szczecin, Żołnierska Str. 48, 71-210 Szczecin, Poland

**Keywords:** Problematic Internet Use, Internet addiction, Internet usage pattern

## Abstract

Problematic Internet Use (PIU) is a broad term that covers problems with Internet use that result in psychological, social, academic or professional difficulties. The aim of our study was to identify individuals particularly vulnerable to developing PIU among Polish students, taking into account sociodemographic characteristics and Internet use patterns. A total of 1008 students of Polish universities took part in the survey. The research tool was The Problematic Internet Use Test—the Polish version of the Internet Addiction Test. Among the respondents, 10.2% showed signs of PIU—high or very high risk of addiction. Significantly higher levels of PIU were found in males than females, science students than medical and humanities students, and Internet users using a computer, as opposed to students using only a phone/tablet. A decrease in PIU was observed as students improved their assessment of their material situation. The severity of PIU increased significantly as the importance of Internet use for entertainment increased. Our research indicates that it is worth implementing measures to prevent the development of PIU in Poland, e.g., screening educational campaigns, especially for science students. It is also necessary to offer students at lower levels of education, especially the less affluent, forms of leisure time other than computer activity with the use of the Internet.

## 1. Introduction

### 1.1. Increasing Number of Internet Users

According to epidemiological studies, more than 40% of the world’s population used the Internet by 2004 (with the exception of Africa, where only 5% of the population was using the Internet at the time) [1]. In 2019, 54% of the world’s population (4.1 billion people) was using the Internet. The sanitary restrictions on movement constraints and quarantine and isolation, introduced universally from March 2020 in the wake of the COVID-19 pandemic outbreak, have resulted in a surge in Internet use. For many, online connections have become an indispensable tool for working, learning and maintaining contacts on an unprecedented scale. By 2021, the number of users has increased by about 10% (800 million), reaching as much as 63% of the world’s population (4.9 billion people). However, around 2.9 billion people are still offline today, 96% of whom live in developing countries. Efforts are being made to include these people in the digital world [2]. 

### 1.2. The Internet—Numerous Opportunities and Accompanying Threats

According to the United Nations specialised agency for telecommunications, the International Telecommunication Union (ITU), the Internet has long been a source of countless opportunities for personal fulfilment, professional development and value creation ([2], p. 1). These are undoubted advantages to the Internet being used as a tool by the average user. However, there is another group that, due to various factors, uses the Internet in a problematic, or in other words, harmful way. There are multiple factors associated with Problematic Internet Use. Epidemiological studies of various countries around the world have highlighted the association of Internet use by young users (teenagers and young adults up to about 25 years of age) with the following: disruption of parental bonds—Greek sample [3]; negative lifestyles in the form of less sleep and lower physical activity—Qatar sample [4]; impaired nighttime sleep quality and excessive daytime sleepiness—Russian sample [5]; eating disorders—Polish sample [6]; and co-occurrence of psychiatric disorders, primarily anxiety and depression [7,8], as well as low self-esteem and associated destructive behaviors such as self-harm [9]. Other researchers have also suggested various physical health problems, such as heart disorders [10], back and neck pain, finger numbness, headaches, inability to sleep, dry eyes or other vision problems, poor nutrition, poor personal hygiene, weight gain/loss and loss of appetite [11]. It seems that the benefits of the Internet cannot overshadow emerging threats.

### 1.3. Internet Addiction and Problematic Internet Use—Concept, Prevalence

In 1996, the term Internet Addictive Disorder (IDA) first appeared in the work of psychiatrist Godberg [12]. He based the criteria for IDA on the criteria for ‘substance dependence’ from the then-current psychiatric diagnostic classification Diagnostic and Statistical Manual of Mental Disorders—DSM [13]. The addict was expected to experience the following symptoms over a twelve-month period: tolerance, withdrawal, lack of control, relapse, large amounts of time spent online, negative consequences and continuation of use irrespective of problem awareness [12]. The first population-based studies conducted according to these criteria in the 1990s revealed that the addicts include both men and women of different ages (the stereotype of only young men being addicted, so-called technophiles, was broken) [14]. The American Psychiatric Association (APA) [15], in the 4th version of the psychiatric classification DSM-IV (also present in the latest version of DSM-V), included in its research appendix the term ‘Internet Gaming Disorder’ as a potential new diagnosis and a condition requiring further empirical and clinical research. To date, the concept of Internet addiction has not been introduced into psychiatric classifications, so various terms are used to describe it, including the following: Internet Addiction Disorder, Compulsive Computer Use, Internet Dependency, Pathological Internet Use, Virtual Addiction and Problematic Internet Use (PIU) [16]. 

In our work, we will use the term Problematic Internet Use (PIU) to broadly encompass problems with Internet use that result in difficulties on psychological, social, academic or vocational grounds, with characteristics of addiction [17]. The term PIU is used by many authors both as a term for behavioural disorders associated with Internet use and in the names of various diagnostic tools, e.g., Problematic Internet Use Scale (PIU) [18,19], Problematic Internet Use Questionnaire (PIUQ) [20,21], Problematic Internet Use Scale (PIUS) [22] and Generalized Problematic Internet Use Scale (GPIUS) [23,24].

The prevalence of Internet addiction worldwide varies widely, depending on the geographical location and sociodemographic characteristics of the group. As indicated by the results of the 2022 meta-analysis, the highest prevalence is found in the WHO African Region (34.53%), followed by the Eastern Mediterranean Region at 30.11%, the South-East Asian Region at 17.43% and the Western Pacific Region 13.91%. The lowest values were shown in the Americas Region (11.06%) and Europe (11.06%) [25]. However, in order to see the scale of the PIU problem, it is necessary to combine these figures with those of Internet access for these populations, which, as we have written, is by far the lowest in African countries (rising to 33% Africa in 2021, compared to 87% in Europe and 81% in the Americas). It may be that populations where there is a high prevalence of Internet and therefore problems with Internet use that were recognised at least a decade ago are relatively better at dealing with Internet use than those countries where it is a fairly new problem [2]. A higher prevalence of problematic use was more common in low-income countries. On a global scale, men (17.15%) were more likely to be affected than women (11.60%) [25], although not all research obtained such a result [26,27,28]. The prevalence of Internet addiction is increasing year by year [25], and mainly affects young people [26,29,30,31,32]. 

### 1.4. The Need to Identify Populations at Particular Risk of PIU in Order to Take Preventive Measures

Unfortunately, the Internet for some groups, especially young people, can become a greater or lesser threat to both mental and physical health, causing difficulties in self-development as well as personal contacts. Given the prevalence of Internet use, including for study or work, diagnosing the problem becomes an important task, that is, identifying those at risk of developing PIU and taking countermeasures.

To date, there is no identified homogeneous group particularly at risk of developing PIU due to gender, level and field of education, material status or patterns of Internet use. This is probably due to cross-cultural differences [33,34], and we therefore felt that research on this topic should be conducted in Poland. Young adults were selected as the target group as those particularly vulnerable to developing PIU.

### 1.5. Purpose of the Study and Hypotheses

The aim of our study was to identify individuals particularly vulnerable to the development of PIU among Polish students by taking into account sociodemographic characteristics (gender, field of study, additional activity outside of studying, assessment of material situation, average number of hours of leisure time per day) and patterns of Internet use (method of Internet connection, main purpose of Internet use, average number of hours spent per day on the Internet due to study/work and other purposes).

In order to achieve the above objectives, the hypothesis was established as follows:

There are relationships between PIU and sociodemographic characteristics and Internet usage patterns among Polish students.

In addition, hypotheses regarding sociodemographic variables were established as follows:Students of different sexes differ in PIU;Students of different fields of study differ in PIU;Students who undertake additional activities outside of studying differ in PIU from students who do not undertake such activities;Students differ in PIU according to self-assessment of material situation;Students differ in PIU according to their average number of hours of leisure time per day.

In terms of Internet use patterns, hypotheses have been established as follows:Students differ in PIU according to the method of Internet connection;Students differ in PIU according to the main purpose of their Internet use;Students differ in PIU according to the average number of hours spent per day on the Internet due to study/work and other purposes.

## 2. Materials and Methods

### 2.1. Study Procedure

The study was cross-sectional, as we indicated in our previous publication [6].The execution of the diagnostic survey was commissioned to an external research company, which conducted the survey in November and December 2018 according to the detailed guidelines of the study’s authors. The premise of the study was to survey students from three public universities in different cities: one with a medical focus, one with a humanities and social sciences focus and one with a technical focus.

Students from the following universities were included in the survey:Medical University of Lublin (central-eastern Poland).University of Economics in Katowice (southern Poland).Silesian University of Technology in Gliwice (southern Poland).

All of the above universities are public.

Once the universities were selected, appropriately trained interviewers contacted lecturers teaching first-, second- and third-year undergraduate students and first-, second and third-year single cycle students. After obtaining permission from the lecturers, the interviewers conducted the survey at the end of the classes. Students were given printed survey questionnaires. Students who did not want to participate in the survey were allowed to leave the room. Before participating in the study, each participant gave informed consent and was informed about the purpose of the study and the right to withdraw at each stage. 

The inclusion criteria for the study were related to the mode of study (I–III year of undergraduate studies and I–III year of single cycle studies), the field of study (medical and health sciences, humanities and social sciences, technical faculties) and an equal number of men and women at each university. Full details on the inclusion criteria were described in detail in our previous paper [6].

The study was approved by the Bioethics Committee of the Pomeranian Medical University in Szczecin (Decision No: KB-0012/188/05/17).

### 2.2. Research Sample

The study involved 1008 Polish university students aged 18 to 40 years (SD = 2.65), with an average age of 21.3 years. The study participants included 510 females (50.6%) and 498 males (49.4%) (Table 1). The study participants were matched by their field of single cycle studies as follows: Medical and health sciences students: 336 people: 174 women and 162 men;Humanities and social sciences students: 336 people: 168 women and 168 men;Students of technical faculties: 336 people: 168 women and 168 men.

There were no statistically significant differences between the number of women and men in the different groups of study fields.

A table with sociodemographic data of the study participants by gender and detailed inclusion criteria is included in our previous publication [6]. Using the EPI Infoprogram^TM^7, the sample size was calculated as 663 people. Based on the available literature [35,36], the prevalence of PIU was estimated to be 10%. The number of full-time students in Poland was 895.725 [37]. The confidence level was 99.0%, and confidence limits were 5%.

### 2.3. Measures

#### 2.3.1. The Problematic Internet Use Test TPUI22 (PIU)

The scale we used in our study [38] is a Polish adaptation of the Internet Addiction Test (IAT) created by Dr Kimberly Young [39]—a pioneer in Internet addiction research. The IAT is a 20-item self-report scale that assesses Internet addiction based on two criteria: ‘substance dependence’ and ‘pathological gambling addiction’. Criteria include loss of control, neglecting everyday life, relationships and alternative recreation activities, behavioural and cognitive salience, negative consequences, escapism/mood modification and deception. The internal consistency of the IAT was determined to be satisfactory, with a Cronbach’s alpha of 0.84 [16]. The scale has so far been adapted into many languages and research has been conducted with it worldwide, e.g., a Korean version [40], Turkish version [41], Italian version [42] and Chinese research [43,44,45]. The Polish version of the IAT by Ryszard Poprawa was called the Problematic Internet Use Test (TPUI22) [38]. The Polish version of the IAT is a valid instrument for measuring Internet addiction [46]. The TPUI22 questionnaire contains 22 questions to which respondents answer on a scale from 0 to 5, where 0 means ‘never’ and 5 means ‘always’. A score between 0 and 110 can be obtained on the test, with a higher score indicating increased problematic use of the Internet. The obtained score can be classified into one of five score categories: very low risk of Internet addiction, low risk of Internet addiction, moderate risk of Internet addiction, high risk of Internet addiction and very high risk of Internet addiction. In the analyses conducted, the high risk of Internet addiction and the very high risk of Internet addiction categories were used as the cut-off point for Problematic Internet Use, e.g., according to the Polish version of the questionnaire, the sum of points 50 and above for persons aged ≤24 years, and 42 and above for persons aged >24 years. Cronbach’s alpha is 0.935 [38]. 

#### 2.3.2. A Self-Designed Socio-Demographic and Internet Use Survey

Our original questionnaire included questions on gender, age, type of study (medical and health sciences; humanities and social sciences; sciences/technical), extracurricular activity (job, volunteering, permanent care of a family member, activity in a student organisation), self-reported financial status (very good, satisfactory, bad) and amount of free time during the week and at the weekend. Regarding the characteristics of Internet use, questions were asked as follows: the main way of Internet use (computer, phone/tablet, computer and phone/tablet to a similar extent), the main purpose of using the Internet (learning, job, entertainment, communication with other people, social media, other, e.g., shopping or online banking) and average number of hours spent daily on the Internet due to study/work and due to other purposes, with a distinction being made between weekdays and weekends.

### 2.4. Methods of Data Analysis

Statistical analysis was carried out in the licensed package IBM SPSS Statistics v.25. Statistical description techniques used were the Kolmogorov–Smirnov test to assess the normality of the distribution of quantitative characteristics, Pearson’s chi-square test for one and two variables, the non-parametric Mann–Whiteney U test for two independent groups and the Kruskal–Wallis test for k-samples, Spearman’s rank correlation and the stepwise linear regression model. A *p*-value < 0.05 was taken as an indicator of statistical significance, while a *p*-value < 0.1 was taken as an indicator of a not fully significant statistical trend.

## 3. Results

### 3.1. Prevalence of PIU

The vast majority of participants (70.5%) had an average risk of Internet addiction (persons without PIU/persons no-PIU), and a total of one in ten (10.2%) showed signs of high or very high risk (persons with PIU: cut-off point of 42 points for persons over 24 years of age and 50 points for persons under 24 years of age) [36] (Table 2).

In the quantitative measurement of PIU, the students surveyed scored between 0 and 110 (M = 25.84; SD = 20.35). The most common score (dominant) was 18 points, and the distribution of scores across the study group was significantly different from normal (*p* < 0.001), with 60.8% of the group scoring below average. Although respondents scored between 0 and 5 on all questions, the highest scoring statement was item No. 1, “I find I’ve been online longer than I intended,” for which the dominant score was 4 (Me = 3.0; M = 2.79; SD = 1.43), followed by item No. 2, “I neglect household chores to spend more time online,” although respondents most often assigned this statement 1 out of 5 points (Me = 2.0; M = 1.96; SD = 1.39). In contrast, the lowest scoring PIU indicators included item No. 14, “I feel preoccupied with the Internet when I’m offline and fantasize about being online” (Me = 0.0; M = 0.62; SD = 1.18), and item No. 12, “I lose my temper or yell when someone disturbs me when I’m online” (Me = 0.0; M = 0.63; SD = 1.18) (see Table 3). 

### 3.2. PIU Score vs. Sociodemographic Variables and Internet Use Characteristics

Both the categorical score and the quantitative score of the PIU measure were differentiated by a number of sociodemographic variables examined. Table 4 provides the results of the frequency comparison of the PIU categories according to the selected qualitative variables. Information on significant correlations was supplemented by comparisons based on the quantitative outcome of the PIU measure.

Men had significantly higher PIU severity, reaching an average of 29.95 measurement points (SD = 24.34), while women reached an average of 21.83 points with a deviation of 14.42 (Z = −4.285; *p* < 0.001). Medical and humanities students were characterised by similar PIU severity, achieving an average of 22.54 points (SD = 15.54) and 22.78 points (SD = 14.59), while science students had a significantly higher severity, achieving an average of 32.20 points with a deviation of 27.01 [H(2) = 15.780; *p* < 0.001]. Among the types of additional activity undertaken outside of studying, considering the categorical score, only volunteering and caregiving significantly differentiated PIU intensity, although analysis of the quantitative results did not confirm these differences—mean PIU intensity was similar among volunteer and non-volunteer practitioners (Z = −1.261; *p* = 0.207) and among caregivers and non-caregivers (Z = −0.709; *p* = 0.478). The mode of Internet use differentiated both categorical and quantitative PIU scores. The highest mean severity was found among those using the Internet via computer (M = 35.96; SD = 29.73), followed by those combining computer and mobile device use (M = 25.47; SD = 21.08), and the lowest PIU severity was found among students using the Internet exclusively via phone/tablet (M = 23.83; SD = 15.81), who constituted the largest of the subgroups [H(2) = 10.68; *p* = 0.005]. 

Relationships between PIU and continuous variables were analysed using the pairwise correlation method, and the results of these analyses are included in Table 5. 

We observed a weak increase in PIU severity with increasing age (rho = 0.066; *p* = 0.037) as well as a decrease in PIU with improving assessment of one’s financial situation (rho = 0.116; *p* < 0.001), suggesting that older and less affluent students may be slightly more prone to Internet overuse. The generalized (regardless of purpose) mean number of hours spent online was also positively associated with PIU, although the relationship was slightly stronger with the mean number of hours spent online Monday to Friday (rho = 0.142; *p* < 0.001) than on weekends (rho = 0.101; *p* = 0.001). However, after taking into account the reason for Internet use, it was observed that correlations only occurred between PIU and time spent online for purposes other than work and study, which was predominantly a source of entertainment and socializing. Internet use for work/study remained independent of PIU, while correlations between PIU and time spent online for other purposes were of similar strength for both the number of hours spent on online entertainment during the week (rho = 0.188; *p* < 0.001) and on weekends (rho = 0.192; *p* < 0.001). Table 4 also shows intercorrelations between the individual indicators, which provide complementary data to the overview of Internet overuse. For example, it turns out that the total number of hours spent online during the week also increases with age (rho = 0.240; *p* < 0.001), but the amount of online time on weekends decreases (rho = −0.076; *p* = 0.016). The amount of time using the Internet for study/work also decreases with age—weaker during the week (rho = −0.087; *p* = 0.005), slightly stronger on weekends (rho = −0.193; *p* < 0.001). The amount of weekend time using the web for entertainment decreases most strongly with age (rho = −0.213; *p* < 0.001). As one’s financial situation improves, the number of hours spent online for entertainment decreases, both during the week (rho = −0.136; *p* < 0.001) and on weekends (rho = −0.144; *p* < 0.001). The average number of hours spent using the Internet entered into multiple relationships with different strengths, as denoted in Table 4.

In addition, several very weak relationships were observed between PIU severity and dominant Internet use goals. PIU severity increased significantly as the importance of Internet use for entertainment increased (rho = 0.097; n = 0.002), while those for whom the dominant purpose of Internet use is learning showed a tendency for PIU severity to decrease (rho = −0.059; *p* = 0.062). 

Finally, a stepwise linear regression analysis was performed using previously correlated quantitative variables. Although the model explained a relatively small percentage of variance [R = 0.307; R^2^ = 0.094; F(4) = 26.009; *p* < 0.001], four variables turned out to be significant predictors of PIU:Along with the increase in time spent on the Internet for purposes other than study/work during the weekend by 1 h, an increase in PIU by 0.25 of the measurement point was observed (t = 8.174; *p* < 0.001);With the increase of age by 1 year, the PIU increased by 0.12 of the measurement point (t = 3.772; *p* < 0.001);The improvement of the financial situation by 1 measurement point resulted in a decrease in PIU by 0.09 points (t = 3.060; *p* = 0.002);A comparable increase in PIU (beta = 0.09) foreshadowed an increase in the weight of Internet use for social network use (t = 3.020; *p* = 0.003).

## 4. Discussion

### 4.1. PIU vs. Age and Gender, Material Status

In our study, every tenth student showed symptoms of PIU. This result is in line with some reports by other researchers before the COVID-19 pandemic, e.g., in Turkey (10.1%) [35] and France (10.2%) [36], although in many countries a lower result of about 8.4% (a range from 1.6% to 12.6%) was obtained [34]. In the first months of the pandemic, the results obtained in Europe were much higher, e.g., 45.1% of students in Lithuania reported PIU [47], and the score among young Italian adults was 33.9% [48]. Interestingly, a higher severity of PIU was shown in COVID-19-infected, -quarantined and -home-isolated individuals [48]. Such a drastic increase in PIU was not observed everywhere. In Hungary, at the beginning of the pandemic, among e-Sport players, e.g., people with a potentially at higher risk of Internet addiction, PIU was shown in 19.9% of those tested [26], while in Mexico and Spain it was shown in about 11.75% [27] and in India it was shown in about 10.54% [32]. In Poland’s neighbors—the Czech Republic and Slovakia—even during the pandemic, lower results were obtained than in our own study—3.5% and 6.2%, respectively, of those at a high or very high risk of PIU [49]. Lower results were also obtained, such as 4.5% among German patients with atopic dermatitis (during the pandemic)—but these were from older age groups (average age 49.9) [50]. The prevalence of PIU therefore varies between populations, even within Europe. The pooled prevalence of Internet addiction worldwide in the general population is 14.22%. The highest percentage is observed in Africa (34.53%), with Europe having the lowest values (11.06%) [25]. Higher results than in Europe are observed in Asian countries [33,34].

In our study, men showed higher PIU severity, achieving higher PIU scores than women. This relationship has also been shown in other studies [3,27,31,32,35,43,51,52,53,54,55,56,57], although this is not always the case [26,27,28]. As already indicated in our previous article [6], this may be related to the purpose of Internet use [53,58,59]—women are more likely to use the Internet to communicate with others, which is less frequently associated with addiction than, for example, use for entertainment.

In the study group (aged 18–40), the propensity to overuse the Internet increased with age. This result is difficult to explain clearly, given that the total number of hours spent online during the week also increases with age, but the amount of time spent online on weekends for entertainment decreases—which may indicate a desire to spend leisure time offline. Interestingly, Kósa et al. obtained opposite results, indicating a higher intensity of PIU among 18–25 year olds than in the 26–35- and 36–45-year-old groups [26]. The trend towards lower age in those at higher risk of PIU is also described in other publications [29,30,31,32]. These differences in the results obtained in our own and other authors’ studies may be explained by the fact that all our subjects were students, and it is university students who are identified as the group most at risk [27,60]. The literature also indicates inter-state differences in the age of those at risk, e.g., for Mexican students it was 20 years and under, and for Spanish students it was 21–35 years [27].

The severity of PIU increased with a lower self-assessment of one’s financial situation, meaning that less affluent students may be slightly more likely to overuse the Internet. At the same time, the number of hours spent online for entertainment both during the week and on weekends decreases as one’s material situation improves. It may be that more affluent students choose leisure activities other than spending time on the Internet, such as going to the cinema, fitness clubs, etc. Our previous research has shown that adolescents, given the opportunity, would choose leisure activities other than spending time online [61]. However, the relationship between socioeconomic status and PIU severity is not a set rule [31,52,62]. 

### 4.2. PIU and the Field of Study and Additional Activity

In our study, science/technical students were more exposed to PIU than medical and humanities students. This is in line with the results of Aznar–Díaz et al. who indicate that science students are more exposed than health sciences [27]. In contrast, in a study by Araba et al., there were more students among education faculty with Problematic Internet Use than medical faculty students [63]. Taking up a job outside of studies was not shown to significantly differentiate PIU severity. However, the results of Kósa et al. indicate that casual work is associated with a higher risk of PIU [26].

### 4.3. PIU and Mode, Purpose, Timing of Internet Use

Balhara et al., in a 2019 publication, indicate that Internet Usage Pattern is more important in explaining Problematic Internet Use than sociodemographic variables [34]. Our research showed that a higher intensity of PIU was characterised by those who used the Internet mainly via computer, while the lowest intensity was found among those who used Internet mainly via phone/tablet. Conversely, among Spanish and Mexican students most at risk of PIU, the tablet was the main device used to connect to the Internet [27].

Those spending time online mainly for entertainment had a higher risk of PIU. Similar results were obtained among Mexican [27], Egyptian [31] and Indian [52] students, and other studies point to social networking, dating and pornography as the main purposes of Internet use associated with PIU [34]. In our research, those for whom the predominant purpose of web use is study/work showed a trend towards lower PIU severity, similar to the research of Ahmed et al. [52]. Other research suggests that using the Internet to search for information is not associated with a greater severity of Internet addiction [31]. Research published to date has already shown a correlation between time spent online and Internet addiction [26,31,34,51,52,57]. Our results indicate that a significant predictor of PIU is increased time spent online for purposes other than studying/working during the weekend. However, it is important to highlight the strong cultural variation in Internet use patterns suggested in other studies [33,34]. 

### 4.4. The Limitations and Prospects of the Study

Our study belonged to cross-sectional research, which entails some limitations. First, the universities were selected by random sampling and appeared to be located only in the southern or central-eastern part of Poland (Gliwice, Katowice, Lublin) [6]. Poland is a fairly homogeneous country in terms of socio-economic development, especially in university cities. There are some differences between east and west, north and south (slightly richer regions are in the west and south of Poland, but also in central Poland) [64]. It should be noted that in the cities where the participants studied, the level of development is comparable. Moreover, the inclusion criterion was the type of field of study, not the location of the university. In the future, when planning studies on the entire population, stratified sampling in four parts of Poland (north, south, east, west) may be considered. 

The second limitation of the cross-sectional nature of the study is the inability to identify risk factors for the development of PIU. For this purpose, a prospective study would have to be conducted.

Future research should take into account the last social changes. As a result of the COVID-19 pandemic, many life activities such as work, study or even social meetings have been moved online, which may contribute to the development of PIU. The war in Ukraine caused an influx of many immigrants to Poland. Ukrainian citizens, far from home and without knowledge of the Polish language, may feel isolated and lonely, which may also increase the risk of PIU. Thus, it seems that it would be advisable to conduct a comparative study of the prevalence of PIU of Ukrainian refugees settled in Poland with Poles.

## 5. Conclusions

The results indicate that the hypothesis stating that there are dependencies between PIU and socio-demographic characteristics and Internet use patterns among Polish students is true.

Taking into account socio-demographic variables, there are dependencies between sex, field of study and self-assessment of the financial situation with PIU. Symptoms of PIU are more common in males than in females, among technical students than among humanities/social science and medical students, and in students with a lower assessment of their material situation (the persons more likely to exhibit PIU symptoms). Sociodemographic variables such as engaging in additional activities outside of studying or the number of hours of free time per day were not shown to differentiate students from PIU. 

Considering the characteristics of Internet use, there are correlations between the way of Internet use (computer/phone), the purpose of Internet use and the number of hours spent online with PIU. Symptoms of PIU are more severe in those who use the Internet via the computer rather than mobile devices, in those who use the Internet mainly for entertainment rather than other purposes and in students who spend more time online, especially for purposes other than studying and working.

Our research shows that PIU is a relatively common phenomenon; therefore, it should be monitored and preventive actions should be taken. An example of such activities may be educational campaigns which provide information about the harmful effects of PIU, actions aimed at increasing psychosocial competences and promoting offline forms of leisure activities, especially on weekends. These could be, for example, sports clubs, hiking clubs, interest circles active in the academic/school environment, systematic development of sports and tourist infrastructure. There also seems to be an urgent need to implement screening among students coming from the group more likely to exhibit symptoms of PIU identified in our study, in order to raise awareness, undertake self-regulation activities and, if necessary, undertake therapy.

According to the obtained results, the recipients of these campaigns should be primarily men, who are more likely to develop PIU than women. This means that educational campaigns or proposals for spending free time in ways other than online should take into account the interests of boys and men. Particular attention should be paid to disseminating preventive measures among students of technical studies. The obtained results showed that PIU is associated with a worse self-assessment of financial situation. Therefore, poorer students should be provided with co-financing for forms of spending time other than online, e.g., discounts on gym memberships, theater tickets and expansion of publicly available sports infrastructure. In our research, PIU was also associated with spending more time online, mainly for purposes other than work and study. Therefore, it is worth introducing self-control training.

To sum up, based on the obtained research results, it should be stated that preventive and screening activities regarding PIU should especially cover men, students of technical faculties and students with a worse self-assessment of their financial situation.

## Figures and Tables

**Table 1 ijerph-20-02434-t001:** Sociodemographics and Internet use characteristics.

Variables	n/M	%/SD	*p **
Gender			
women	510	50.6	0.729
men	498	49.4	
Age:(median = 21.0)	21.31	2.65	<0.001
Course type			
Medical and health sciences	336	33.3	1000
Humanities/social sciences	336	33.3	
Science/technical	336	33.3	
Additional activity outside of studying			
job	515	51.1	0.508
voluntary work	130	12.9	<0.001
permanent care of a family member	93	9.2	<0.001
activity in a student organisation	110	10.9	<0.001
none of the above	391	38.9	<0.001
Assessment of the financial situation			
very good	297	29.5	<0.001
satisfactory	662	65.7	
bad	49	4.8	
Average number of hours of leisure time per day			
during the week (Mon-Fri)	2.91	2.08	<0.001
at weekends	6.06	3.60	<0.001
Main way of Internet use			
computer	108	10.7	<0.001
mobile phone/tablet	462	45.8	
computer and phone/tablet to a similar extent	438	43.5	
The main purpose of using the Internet			
learning	284	28.3	<0.001
job	66	8.2	<0.001
entertainment	161	16.0	<0.001
communication with other people	356	35.5	<0.001
social media	128	12.8	<0.001
other, e.g., shopping or online banking	50	5.0	<0.001
Average number of hours spent daily on the Internet due to study/work			
during the week (Mon-Fri)	3.17	5.06	<0.001
at weekends	3.08	2.06	<0.001
Average number of hours spent online per day for other purposes			
during the week (Mon-Fri)	2.95	3.35	<0.001
at weekends	4.03	3.70	<0.001

* Significance levels given indicate the symmetry of the distributions, obtained by the one-sample chi-square test (for qualitative variables) and the Kolmogorov–Smirnov test (for quantitative variables).

**Table 2 ijerph-20-02434-t002:** PIU measurement results—risk categories of Internet addiction.

Variables	n	%
PIU		
very low	33	3.3
low	161	16.0
average	711	70.5
high	64	6.3
very high	39	3.9

**Table 3 ijerph-20-02434-t003:** Respondents’ responses to individual questions on the TPUI22 questionnaire.

Item	Mode	Me	M	SD
1. I find I’ve been online longer than I intended.	4	3	2.79	1.43
2. I neglect household chores to spend more time online.	1	2	1.96	1.39
3. I prefer the excitement of the Internet over the closeness of my partner, friends or family.	0	1	1.02	1.28
4. I create new relationships with fellow online users at the expense of relationships with people offline.	0	0	0.68	1.21
5. People around me complain about the amount of time I spend online.	0	1	1.17	1.38
6. I happen to say “just a few more minutes” when I’m online.	1	1	1.87	1.53
7. I compromise my performance at work/school because of too much time spent online.	1	1	1.63	1.42
8. I happen to hide what I’m really doing online when asked about it.	0	1	1.09	1.32
9. Going online calms down troubling thoughts about my life.	0	1	1.33	1.51
10. I realize that I am thinking about when I will be online again.	0	0	0.94	1.33
11. I worry that my life without the Internet would be boring, empty and joyless.	0	1	1.24	1.41
12. I lose my temper or yell when someone disturbs me when I’m online.	0	0	0.63	1.18
13. I sometimes neglect my sleep due to being online for long periods of time.	1	1	1.55	1.44
14. I feel preoccupied with the Internet when I’m offline and fantasize about being online.	0	0	0.62	1.18
15. My performance at work/school suffers because I spend too much time online.	0	1	1.19	1.37
16. I’ve tried to reduce the amount of my online sessions without success.	0	1	1.07	1.26
17. I try to hide from others how long I’ve been online.	0	0	0.74	1.25
18. I choose to spend time online instead of hanging out with friends or family.	0	0	0.79	1.23
19. I feel irritated/moody, nervous or depressed when I’m offline, but these feelings disappear when I go back online.	0	0	0.71	1.24
20. Being online helps me relieve my negative feelings (e.g., hopelessness, sadness, depression, anxiety or guilt).	0	1	1.20	1.39
21. I feel anxious, annoyed or depressed at the thought of having to limit my Internet use.	0	0	0.91	1.32
22. I notice that I need to increase the time I spend online to achieve satisfaction from using the Internet.	0	0	0.71	1.26

**Table 4 ijerph-20-02434-t004:** Comparison of PIU risk category groups by examined variables.

Variables	df	Chi^2^	*p*
Gender	4	54.318	<0.001
Course type	8	80,650	<0.001
Additional activity outside of studying:			
job	4	6.293	0.178
voluntary work	4	26.241	<0.001
permanent care of a family member	4	41.368	<0.001
activity in a student organisation	4	2.302	0.680
none of the above	4	7760	0.101
Main way of Internet use	8	79,842	<0.001

**Table 5 ijerph-20-02434-t005:** Spearman’s correlations between the severity of PIU and selected factors.

Variables	1	2	3	4	5	6	7	8
1. PIU	-							
2. Age	0.066 *	-						
3. Financial situation	−0.116 **	−0.028	-					
4. Average number of hours spent online during the week	0.142 **	0.240 **	0.004	-				
5. Average number of hours spent online at weekends	0.101 **	−0.076 *	−0.076 *	0.425 **	-			
6. Average time of using the Internet for study/work during the week	−0.036	−0.087 *	0.020	0.005	0.009	-		
7. Average time of using the Internet for study/work at weekends	0.042	−0.193 **	0.056	0.032	−0.048	0.640 **	-	
8. Average time of using the Internet for other purposes during the week	0.188 **	−0.037	−0.136 **	0.282 **	0.206 **	0.280 **	0.228 **	-
9. Average time of using the Internet for other purposes at weekends	0.192 **	−0.213 **	−0.144 **	0.271 **	0.36 **	0.252 **	0.267 **	0.712 **

* *p* < 0.05; ** *p* < 0.01.

## Data Availability

The data presented in this study are available on request from the corresponding author.

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
