# Peer review of "Problematic Internet Use among Polish Students: Prevalence, Relationship to Sociodemographic Data and Internet Usage Patterns"

_ijerph, 2023, doi:10.3390/ijerph20032434_

Round 1

Reviewer 1 Report

In the post-pandemic era in which we live, everything points to increased internet and social media use by children, adolescents, young people and adults. Despite the multiple benefits these technologies provide, as research has shown, they may also negatively affect people's well-being. Therefore, more research should be done on this issue and, above all, on how to prevent it. The study is interesting, but several aspects need to be corrected in detail by the authors:

- What were the main starting hypotheses that guided the research? These must be explicitly stated in the study.

- In the methodology, it is necessary to add a specific section on "Procedure". Here, the authors should explain, in detail, how the study was conducted. For example, how the sample was selected and accessed. In the same way, how the instruments were administered and how the research process was carried out.

- It would be necessary to contextualise the sample universities: were they public or private, in which areas of the country were they located, and why were these universities chosen?

- Where does the following information come from? "... total of one in ten (10.2%) showed signs of 205 high or very high risk (persons with PIU: cut-off point of 42 points for persons over 24 206 years of age and 50 points for persons under 24 years of age).

- Add the year of publication of Balhara et al.

- Authors are encouraged to include all items in the questionnaire used in the study (Adapted problematic internet use test), as this could be useful for other studies.

- The conclusion is quite general. What are the conclusions based on the hypotheses?

- In the conclusion section, the authors point out that the study can help develop educational campaigns or leisure activities at weekends. They also refer that actions can be carried out at lower educational levels, such as sports clubs. It would be essential to review the contributions of the study. For example, prevention could be approached by considering the self-regulation of using these technologies by students.

- Add to the theoretical framework the studies of Caplan, S. E. (2002, 2007, 2010), who has researched problematic internet use.

- Add the limitations and prospects of the study.

Author Response

Dear Reviewer,

Thank you very much for all Your comments. Below we list all the changes made in response to the review, which are also included in the manuscript using the "Track Changes" function. In addition, we have made minor changes to the vocabulary also with the function using the "Track Changes" function, too.

Kind regards,

---

Authors

Reviewer 2 Report

Introduction and theoretical framework:

They reflect very well what are the starting points of this research, taking into account relevant references and setting very well what are the variables of study, such as the skills and expectations of these students

Material, methods and methodology:

The methodology, as well as the analyses carried out are those that correspond to this type of study. 

Typing errors should be corrected in table 1 "):" under "course type".

 Results: The results presented faithfully reflect, both in the tables and in the graphs, the confirmation of the hypothesis and the existing relationship between the different variables.

 Discussion of the results and conclusions:

The discussion of the results is correctly adjusted to the analyses performed, as well as a good comparison is made with other studies and authors, which allows us to see the reliability and robustness of the results.

At the same time, the future prospects and future lines of research, as well as the limitations of the study must be presented.

 References: Both the references that are presented are adjusted to the subject, and at the same time it should be noted that they are the correct ones to validate the results.

Author Response

(The authors gave the same response as above.)
